# Research on feature selection for AC contactor vibration signals based on regularized random forest with recursive selection

Shuxin Liu⊚, Xinzhi Qi⊚ⵔ*, Chaojian Xing, Xin Ming⊚, Xianfeng Lv

Key Laboratory of Special Electric Machines and High Voltage Apparatus in the Ministry of Education, Shenyang University of Technology, Shenyang, China

⊚ These authors contributed equally to this work.
* Qixinzhi_1999@163.com

## Abstract

When conducting condition recognition research on AC contactor vibration signals through time-frequency analysis, the feature data exhibit a high degree of redundancy, which leads to repetitive information and hinders the accuracy of recognition. To address the redundancy issue in the features of AC contactor vibration signals, this study introduces a feature selection method based on Regularized Random Forest with Recursive Selection (RFRS). Initially, a test platform for AC contactor vibration signals was established, and time-frequency domain features of the AC contactor vibration signals were extracted. Subsequently, the traditional Random Forest (RF) was refined by optimizing its stopping criteria using the Recursive Feature Elimination approach and by incorporating a regularization coefficient during the splitting process to direct the split towards significant features. This modification not only enhances the Random Forest's capacity to leverage existing information but also introduces a bias, enabling it to favor important features. Finally, through case analysis, the proposed method effectively reduced the dimensionality of the feature set and achieved an average of 87.37% for Recall, 87.41% for F1-Score, 88.38% for Precision, and 85.74% for Accuracy. The overall performance of this method surpasses that of the three mainstream feature selection methods: Spearman's rank correlation coefficient method, the embedded method, and the filter method. This study thus provides a rather effective feature selection approach for the state recognition study of AC contactors.

## 1. Introduction

Under the backdrop of the "dual carbon" strategic goals, the construction and development of a new type of power system have become a key pathway to achieving long-term sustainable development [1–3]. The new power system demands higher reliability from control equipment. The AC contactor, which is a control electrical device used for frequently making and breaking the main circuit of motors or loads, plays a crucial role in the safe operation of the

**Data Availability Statement:** Data Availability Statement: The dataset supporting the findings of this study is available in the GitHub repository

named "AC_contactor". The specific file containing the vibration signal data is "AC_contactor_vibration_signals.csv", which can be accessed directly at the following URL: https://github.com/zzzyizhi/AC_contactor/blob/master/AC_contactor_vibration_signals.csv. This link provides a direct and permanent access to the dataset used in our research.

**Funding:** This study was supported by the National Natural Science Foundation of China (No. 51977132). The funder had no role in the study design, data collection and analysis, decision to publish, or preparation of the manuscript.

**Competing interests:** The authors have declared that no competing interests exist.

new power system. Therefore, it is essential to conduct a reliability analysis of the AC contactor, and this analysis largely depends on the selection of key feature data.

Currently, most scholarly research on the state recognition of AC contactors primarily focuses on electrical and mechanical parameters. In reference [4], five parameters including contact resistance and closing time are studied. The remaining electrical life of the contacts is predicted using Pearson's correlation coefficient method and LSTM neural network, achieving an accuracy rate of over 93%. Reference [5] employs the Wiener method to statistically analyze the quality loss data of the contacts at different stages, predicting the remaining life of the contacts with an error of less than 10%. Reference [6] utilizes an autoencoder neural network to compress and integrate electrical and mechanical parameters, extracting a comprehensive health indicator that characterizes the contact state, which enhances robustness by 5% compared to traditional methods. References [7–10] implement predictions of the remaining life of the contacts using various neural network algorithms. Reference [11] pre-processes the electrical and mechanical parameters into high-dimensional data and applies principal component analysis to reduce the dimensionality of this data. Subsequently, based on subjective experience, important components are selected as inputs for PSO-LSSVM to more accurately identify the degradation state of AC contactors on small sample datasets. Reference [12] establishes an audio model of the contactor during the making process and predicts the remaining life of the contactor with a lower error method combined with CNN algorithm. The aforementioned studies concentrate on introducing new algorithms into traditional datasets without conducting a detailed study on the vibration signals of AC contactors, and have yet to determine the key features that best represent the state of the contactors. In contrast, there is a certain consensus among researchers on the use of vibration signals for critical power equipment, such as employing time-domain amplitude and frequency-domain amplitude as key features for identifying the operating state of high-voltage circuit breakers [13]. This consensus aids subsequent researchers in reducing the scale of data and enhancing model recognition accuracy [14, 15]. Therefore, it is of great importance to conduct an in-depth study of the variation patterns of AC contactor vibration signals and to identify the time-domain and frequency-domain features that can best represent the changes in the state of the contactors.

Statistical analysis of vibration signal time series and spectral data can yield feature data such as amplitude, amplitude standard deviation, and amplitude variance. However, there is often a varying degree of correlation among these features; the higher the correlation, the greater the redundancy in information, ultimately leading to data redundancy that reduces both database storage space and the accuracy of condition recognition. Mainstream feature selection methods include the correlation coefficient method, embedded methods, filter methods, and recursive elimination methods. The correlation coefficient method is relatively simple, but the setting of the threshold is subjective and requires multiple subjective judgments to select the optimal features [16], hence it is inefficient on high-dimensional data sets [17–20]. The embedded method, which typically uses the random forest model as a base learner, is suitable for high-dimensional data sets and can greatly improve the efficiency of feature selection compared to the correlation coefficient method. However, due to the model's model's lack of bias, the embedded method may split nodes along non-critical features, thereby affecting the model's decision-making process [21, 22]. Recursive elimination method exhaustively searches all possible feature combinations, which is conducive to finding the true key features, but it often takes a lot of time on high-dimensional data sets. Therefore, this study introduces a regularization coefficient into the random forest splitting process to guide the random forest to split in the direction of important features, and combines the recursive elimination method to optimize the stopping criterion of the random forest, achieving an organic combination of subjective and objective in the feature selection process.

In summary, to further explore the application of vibration signals in the field of state recognition for AC contactors and to address the issue of high redundancy in both time-domain and frequency-domain features of these signals, this study proposes a vibration feature selection method for AC contactors based on RFRS. Firstly, this study constructs an experimental platform for AC contactors based on vibration signals, in accordance with relevant testing standards. Subsequently, the platform is utilized to extract both time-domain and frequency-domain features of the vibration signals. Furthermore, by introducing a regularization term and integrating the concept of recursive feature elimination, improvements are made to the random forest algorithm, enabling it to effectively address the selection of time-frequency domain features from AC contactor vibration signals.

## 2 Fundamental theory

### 2.1 Random forest

RF is a robust ensemble model that enhances overall recognition accuracy by establishing multiple decision trees and aggregating all results. Each feature is considered as a node, and when the input sample passes through these nodes, the decision tree evaluates the importance of each feature based on information gain. The implementation process of RF is shown in Fig 1.

The specific process can be described by the following steps:

(1) Let the dataset be defined as $X = \{(\boldsymbol{x_1},y_1),(\boldsymbol{x_2},y_2),...,(\boldsymbol{x_N},y_N)\}$, where $N$ is the number of samples. For the $j$-th sample $(\boldsymbol{x_j},y_j)$ $(j = 1,2,...,N)$, $y_j$ represents the state class (with a total of $Y$ classes), and $\boldsymbol{x_j}$ is the feature vector. When there are $F$ features, $\boldsymbol{x_j}$ is represented as $(x_{j,1}, x_{j,2},...,x_{j,F})$, where $x_{j,f}(f = 1,2,...,F)$ denotes the value of the $f$-th feature for the $j$-th sample.

(2) Randomly extract $n(1 \leqq n \leqq N)$ subsets of equal size from the dataset $X$. These subsets are used to construct $n$ decision tree models. For the $m$-th decision tree model $(1 \leqq m \leqq n)$, the input dataset is denoted as $X_{\text{subset},m}$, with a sample size of $N_{\text{subset},m}$ and the number of features being $F$.

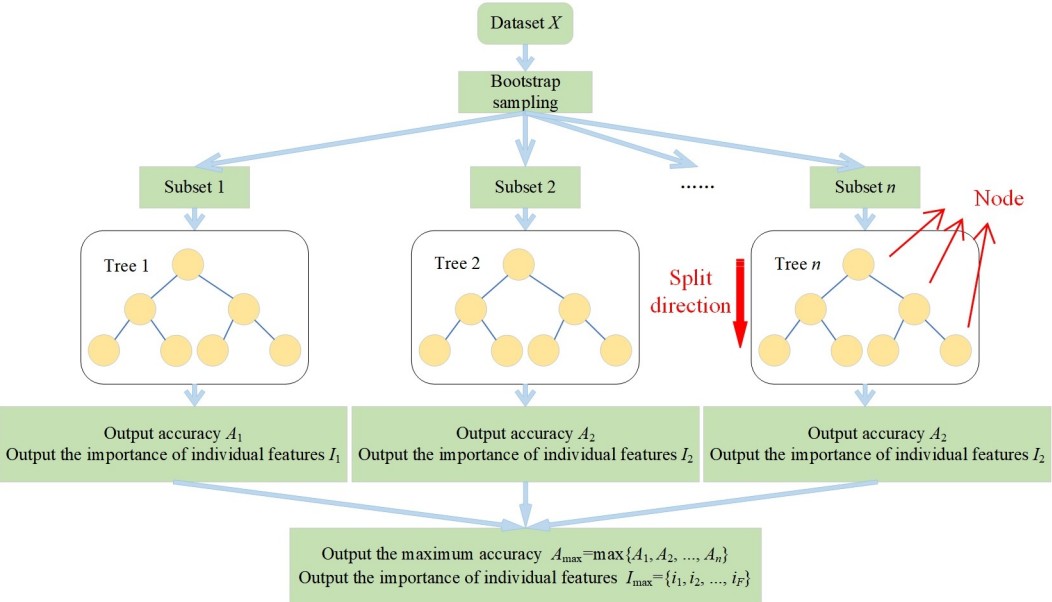

**Fig 1. Principle diagram of RF.**

(3) The decision tree calculates the Gini index ($C_{\text{Gini},v}$) for the $v$-th node based on the samples of different classees at that node. It also calculates the Gini index for the left child node ($C^{L}_{\text{Gini},v+1}$) and the right child node ($C^{R}_{\text{Gini},v+1}$) of the subsequent node, using the difference in Gini index between the $v$-th node and the ($v$+1)-th node as the information gain to determine the direction of the decision tree split. The relevant calculation formulas are as follows:

$$
\begin{cases}
C_{\text{Gini},v} = \sum_{y}^{Y} \left( \frac{N_{\text{subset},m}}{N_y} \right)^2 \\
C_{\text{Gain},v} = C_{\text{Gini},v} - w_{\text{R}} C^{R}_{\text{Gini},v+1} - w_{\text{L}} C^{L}_{\text{Gini},v+1} \\
w_{\text{R}} = \frac{N^{R}_{v+1}}{N_{\text{subset},m}} \\
w_{\text{L}} = \frac{N^{L}_{v+1}}{N_{\text{subset},m}}
\end{cases}
\tag{1}
$$

In Eq (1), $N_y$ represents the number of samples for state $y$ at the $v$-th node, while $N^{R}_{v+1}$ and $N^{L}_{v+1}$ denote the number of samples at the right and left child nodes, respectively, of the ($v$+1)-th node. $w_{\text{R}}$ and $w_{\text{L}}$ represent the proportions of the samples in the right child node $N^{R}_{v+1}$ and the left child node $N^{L}_{v+1}$ relative to $N_{\text{subset},m}$, respectively.

(4) After the node splitting is completed, all decision trees calculate the overall accuracy ($A$) and return the results to RF. The formula for calculating the overall accuracy is as follows:

$$
A_m = \frac{N_{\text{correct},m}}{N_{\text{subset},m}}
\tag{2}
$$

In Eq (2), $N_{\text{correct},m}$ represents the number of samples correctly identified by the $m$-th decision tree model.

(5) The RF selects the decision tree with the highest accuracy rate as the final model and, based on the information gain of this decision tree model, calculates the importance of each feature. The formula for this calculation is as follows:

$$
\begin{cases}
C_{\text{Gain},f} = C_{\text{Gain},v} \\
i_f = \frac{\sum_{f=1}^{F} C_{\text{Gain},f}}{\sum_{m=1}^{n} \sum_{f=1}^{F} C_{\text{Gain},f}}
\end{cases}
\tag{3}
$$

In Eq (3), $i_f$ denotes the importance of the $f$-th feature. $C_{\text{Gini},f}$ represents the information gain of the $f$-th feature. Typically, a node corresponds to a feature, hence the information gain of the $v$-th node is equivalent to the information gain of the $f$-th feature.

## 2.2 Regularized random forest with recursive selection

In previous studies, most researchers have been inclined to introduce regularization terms into the tree structure and feature weights of RF to control the depth and complexity of the tree models, as shown in Eq (4). This approach transforms complex practical problems into tree model construction and optimization issues, enabling the widespread application of RF across

various professional fields. However, this characteristic limits RF's ability to fully utilize known information to control the direction of node splits when calculating feature importance. Consequently, the decision outcomes may be influenced by certain nodes that only offer minimal gains, resulting in the ineffective selection of key features by RF.

$$
\begin{cases}
C_{\text{Gain},v} = C_{\text{Gain},v} & if \quad f \in S \\
C_{\text{Gain},v} = \lambda C_{\text{Gain},v} & if \quad f \notin S
\end{cases}
\tag{4}
$$

In Eq (4), $S$ denotes the optimal set of features for the $v$-th node.

To effectively utilize known information to control the direction of node splitting, this study modifies Eq (4) as follows:

$$
\begin{cases}
C_{\text{Gain},v} = C_{\text{Gain},v} & if \quad f \in S \\
C_{\text{Gain},v} = \alpha_f C_{\text{Gain},v} & if \quad f \notin S
\end{cases}
\tag{5}
$$

Additionally, the calculation formula for $\alpha_f$ is as follows:

$$
\alpha_f = (1 - \gamma)\lambda + \gamma \frac{i_f}{i_{\max}}
\tag{6}
$$

In Eq (6), $i_{\max}$ represents the maximum importance among the $F$ features.

The value of $\alpha_f$ is constituted by two components: a penalty coefficient and feature information. $\gamma$ and $\lambda$ represent two distinct penalty coefficients, each with a value ranging from 0 to 1. Specifically, $\gamma$ is utilized to regulate the influence of the information from the $f$-th feature on the node's information gain. Upon comparing the model before and after modification, it becomes clear that the original model applies $\lambda$ to exert an equal degree of control across all features. In contrast, the modified model employs $\gamma$ to take full advantage of known feature information, thereby enabling differential control over each feature. Additionally, the modified version of Eq (6) retains $\lambda$, and when $\lambda = 0$, the modified model reverts to the original Random Forest model, suitable for addressing tasks beyond feature selection.

It should be considered that the modified model can only more accurately calculate the importance of each feature, and the judgment of key features still necessitates the integration of subjective experience. Therefore, this study further optimizes the model by incorporating the concept of recursive feature elimination. Assuming that the features obtained from the modified model are ordered by importance from highest to lowest as $Z = \{z_1, z_2, \ldots, z_F\}$, the first $k$(Where $k \leqq F$) features of set $Z$ are used as the input feature matrix for the final model. The results of the final model are then assessed twice according to the elbow method. The first assessment uses overall accuracy as the metric, comparing the difference in overall accuracy between the $k$-th and $(k+1)$-th combinations. The purpose of this assessment is to identify features that significantly enhance the identification capability. The second assessment uses the recognition accuracy rates for each state of the AC contactors as the metric, with the calculation formula referring to Eq (7). The significance of this assessment lies in mitigating the impact of outliers in the recognition accuracy rates for individual states. In summary, to effectively address the feature redundancy in AC contactors, this study proposes an RFRS-based feature selection method, which refines the random forest model. The operational principle is detailed in Fig 2.

$$
A_{k,y} = \frac{TP_y + TN_y}{TP_y + TN_y + FP_y + FN_y}
\tag{7}
$$

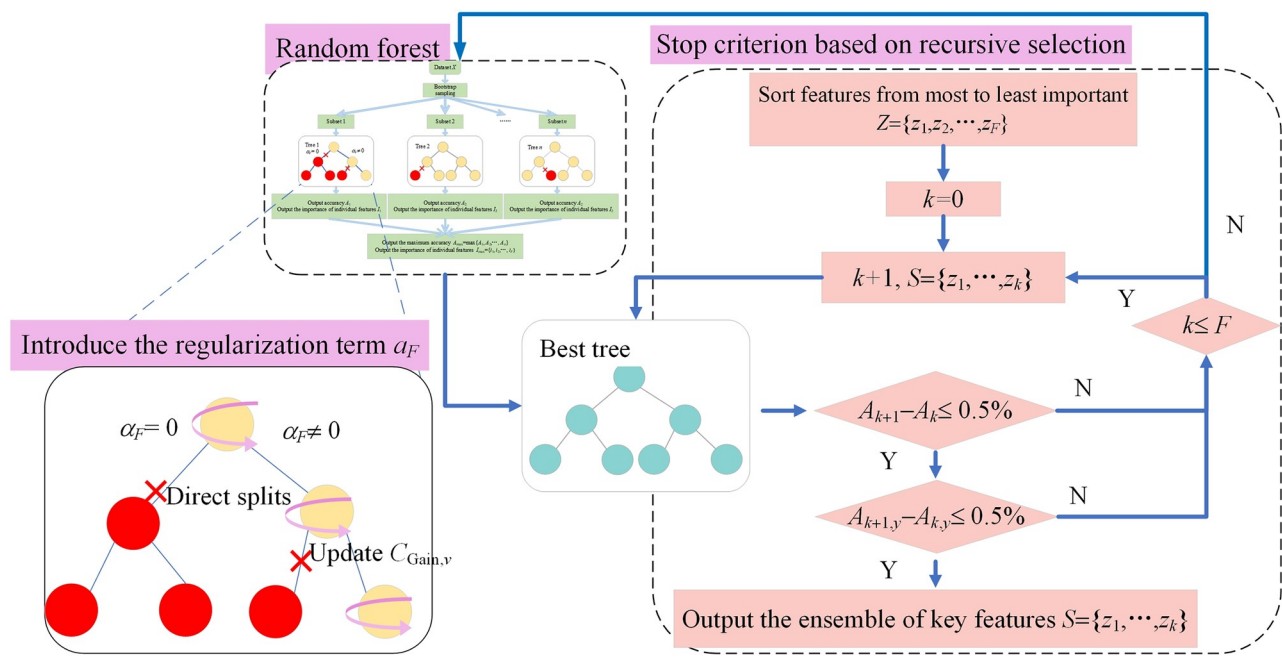

**Fig 2. Schematic of the RFRS implementation principle.**

In Eq (7), $TP_y$ represents the number of samples correctly identified as belonging to class $y$, $TN_y$ refers to the number of instances accurately classified as not belonging to class $y$, $FP_y$ denotes the number of samples that are incorrectly classified as class $y$ when they belong to other class, and $FN_y$ indicates the count of samples from class $y$ that were mistakenly identified as belonging to other class.

## 3 Test process and data acquisition

### 3.1 Test principle

AC contactors rely on the closing and separation actions of their moving and static contacts to manage the conduction and interruption states of the circuit, as depicted in Fig 3. When

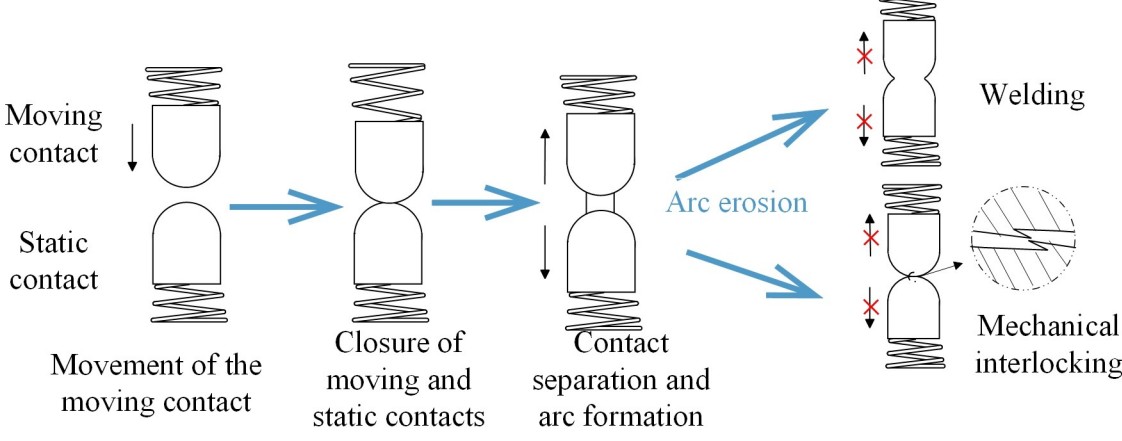

**Fig 3. Schematic diagram of the working principle and fault conditions of an AC contactor.**

current flows through the spring coil, the moving contact is subjected to an electromagnetic repulsive force, causing it to move towards and eventually close with the static contact. At this juncture, the contactor is in a closed state, the circuit is active, and current flows freely. Upon cessation of current through the spring coil, the moving contact, influenced by the spring's pull, disengages from the static contact. At this point, the contactor is open, and the circuit is interrupted. With an increasing number of operations, the erosive effects of electrical arcs can lead to irreversible deformation of the contact surfaces. Such deformation may result in welding or mechanical interlocking of the moving and static contacts, preventing their normal separation and causing the AC contactor to malfunction, which can lead to an inability to interrupt the circuit.

## 3.2 Test conditions

Reliability analysis of AC contactors must adhere to certain standards, meaning that feature data must be obtained under specified test conditions. Therefore, this study constructs a full-life-cycle test platform for AC contactors in accordance with the Chinese National Standard GB14048.4–2010, with some test conditions detailed in Table 1.

Fig 4A illustrates the experimental platform for AC contactor vibration signals constructed in this study. This platform utilizes a KS76C100 vibration sensor to measure vibration signals. The sensor has an acceleration range of 60g and an output signal ratio of 100mV/g, demonstrating good measurement capabilities. Additionally, the sensor can be directly connected to a data acquisition card, which helps to minimize interference from external noise.

When the power console issues a closing command, the moving and static contacts close, generating vibration due to the impact during this closing process. Upon the issuance of an opening command, the moving and static contacts separate, and the spring mechanism connected to the contacts rebounds, leading to vibration. The sensor, mounted on the contactor housing, measures these vibrations at a sampling frequency of 1 MHz/s. The duration of the vibration signal during the opening process can be calculated using Eq (8), and the signal's frequency spectrum is obtained via Fourier transform. Fig 4B illustrates the time-domain and frequency-domain diagrams of the vibration signal for a complete make-and-break operation.

$$\text{Time} = N\big/_{1\times10^6} \tag{8}$$

In Eq (8), $N$ represents the number of sampling points.

## 3.3 Data acquisition

**3.3.1 Feature extraction.** As can be inferred from the fault principles of AC contactors in Section 2.1, welding or mechanical interlocking phenomena can lead to uncertainties in the separation action. Such uncertainties are critical in determining the reliable control of the circuit. Therefore, this study focuses on the vibration signals during the separation process as the

**Table 1. Test conditions.**

| Parameter | Value | Parameter | Value |
|---|---|---|---|
| Coil voltage/V | AC 220 | Work system | AC-4 |
| Load voltage/V | 400(380) | Load current/A | 240 |
| Load type | RL | Power factor | 0.35 |
| Operating frequency/(times/h) | 300 | Sampling frequency/Hz | 1M |

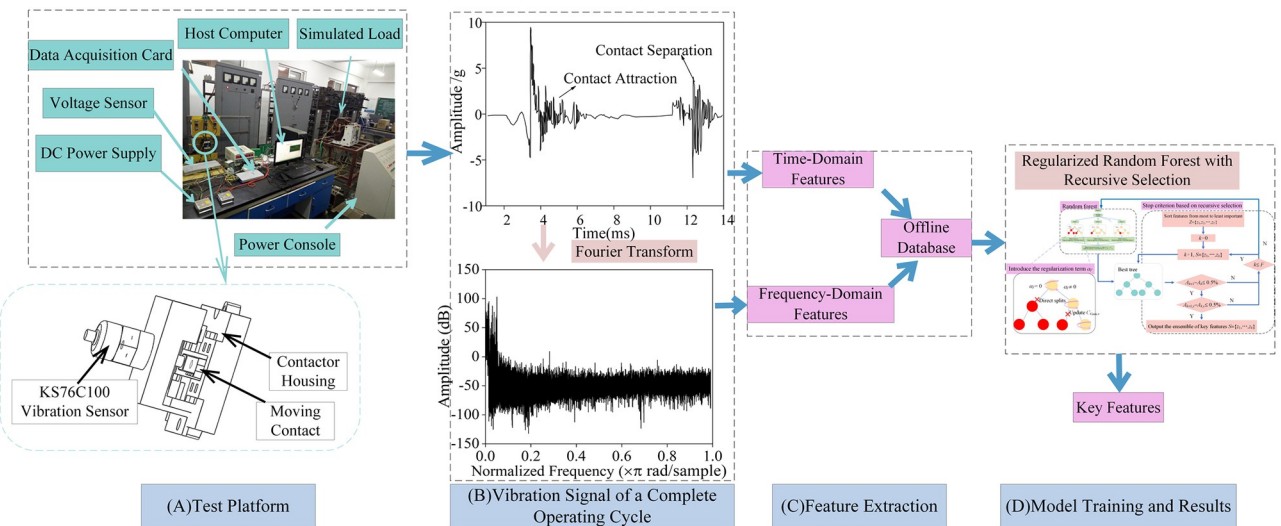

**Fig 4. Test process.** (A)Photograph of the AC contactor test platform and schematic diagram of vibration sensor installation (B)Vibration signal and frequency spectrum of an AC contactor during a single make-and-break operation. (C)Time-frequency domain feature extraction. (D)RFRS training and test results.

subject of research, from which time-frequency domain features are extracted. The extraction method is shown in Table 2.

In Table 2, $x_i$ denotes the amplitude of the $i$-th sample in the time domain, $X_i$ represents the amplitude at the $i$-th frequency point, $f_i$ refers to the frequency-domain value.

**3.3.2 Sample classification.** When samples are entered into the offline database, accurate classification of the samples is crucial for subsequent data analysis or decision-making activities. this study, taking the CJX2-5011 model AC contactor as an example, elaborates on the criteria for sample classification.

In Section 2.1, the working principle and failure mechanisms of AC contactors are described in detail. Research indicates that the failures of AC contactors are primarily divided into two types: welding and mechanical interlocking. Although these two failure phenomena have different manifestations, they are consistent in physical essence (there is a deviation in the relative position of the moving and static contacts). As shown in Fig 3, the kinematic manifestation of the welding phenomenon is that after the moving and static contacts are engaged, the moving contact continues to move forward for a certain distance, a process that can be characterized by overtravel. The kinematic manifestation of the mechanical interlocking phenomenon is that after the moving and static contacts are engaged, they cannot separate normally, a process that can be characterized by the minimum distance between the moving and static contacts (opening distance). By measuring the relative position between the moving and static contacts, the current working state of the contacts can be determined.

As depicted in Fig 5, this study presents the variations in overtravel and opening distance of the contactor throughout its entire service life. The change in opening distance exhibits a certain periodic characteristic. In contrast, the variation in overtravel demonstrates a stepwise pattern, aligning with the irreversible degradation of the contact surface. During Stage I (with the number of operations ranging from 1 to 40,000 times), the amplitude change of overtravel is not significant; after Stage II (with the number of operations ranging from 40,000 to 70,000 times), there is a noticeable increase in the amplitude of overtravel; and in Stage III (with the number of operations ranging from 70,000 to 90,000 times), the rate of increase in overtravel

**Table 2. List of time-frequency domain feature.**

| Name | Formula | Description |
|---|---|---|
| Time-domain peak-to-peak amplitude (T1) | $T1 = \max(x_i) - \min(x_i)$ | The difference between the maximum and minimum signal values within a specified time frame, indicating the signal's range of variation. |
| Time-domain amplitude mean (T2) | $T2 = (\sum_{i=1}^{N} x_i)/N$ | The average value of the signal amplitudes over a specified time interval, reflecting the central tendency of the signal's amplitude. |
| Time-domain amplitude absolute mean (T3) | $T3 = (\sum_{i=1}^{N} |x_i|)/N$ | The mean value of the absolute signal amplitudes over a specified time period, indicating the average energy of the signal. |
| Time-domain amplitude root mean square (T4) | $T4 = \sqrt{(\sum_{i=1}^{N} x_i^2)/N}$ | The square root of the mean of the squares of the signal amplitudes over a specified time period, a measure of the effective value of the signal's amplitude. |
| Time-domain amplitude standard deviation (T5) | $Tk = \sqrt{(\sum_{i=1}^{N} (x_i - T2)^2)/N}$ | A measure of the amount of variation or dispersion of a set of signal amplitudes from their mean value. |
| Kurtosis factor (T6) | $T6 = (T2/T5)^4$ | A statistical measure that describes the tail weight of the amplitude distribution in comparison to a normal distribution, indicating the peakedness or flatness of the signal distribution. |
| Skewness factor (T7) | $T7 = (T2/T5)^3$ | A statistical measure that quantifies the asymmetry of the amplitude distribution around its mean, indicating the direction and degree of the data's departure from normal distribution symmetry. |
| Waveform factor (F1) | $F1 = T4/T3$ | The ratio of the root mean square value to the mean value of the signal amplitude, reflecting the waveform's fluctuation characteristics. |
| Peak factor (F2) | $F2 = T1/T4$ | The ratio of the peak amplitude to the root mean square value of the signal, indicating the relative magnitude of the highest peak in comparison to the average energy of the signal. |
| Impulse factor (F3) | $F3 = (T1/T3)^3$ | A measure of the impulsive characteristics of the signal, typically indicating the presence of short-duration, high-amplitude events. |
| Safety margin factor (F4) | $F4 = T1/(T3)^2$ | A measure of the safety margin or the difference between the system's stability threshold and its current operating point, reflecting the system's resilience to disturbances. |
| Frequency-domain peak-to-peak amplitude (F5) | $F5 = \max(x_i) - \min(x_i)$ | The difference between the maximum and minimum amplitude values in the frequency domain, indicating the range of amplitude variation across frequencies. |
| Frequency-domain amplitude mean (F6) | $F6 = (\sum_{i=1}^{N} X_i)/N$ | The average value of the amplitudes across the frequency spectrum, providing a measure of the central tendency of the signal's energy distribution. |
| Frequency-domain amplitude median (F7) | $F7 = \text{median}(X_i)$ | The median value of the spectral amplitudes when sorted in ascending order, used to mitigate the impact of extreme values on the analysis. |
| Frequency center (F8) | $F8 = (\sum_{i=1}^{N} f_i \times X_i)/(\sum_{i=1}^{N} X_i)$ | The average value of the upper and lower cutoff frequencies, representing the central frequency of the signal's amplitude distribution in the frequency domain. |
| Frequency root mean square (F9) | $F9 = (\sum_{i=1}^{N} f_i^2 \times X_i)/(\sum_{i=1}^{N} X_i)$ | The square root of the mean of the squares of the amplitudes across the frequency spectrum, indicating the effective amplitude level of the signal in the frequency domain. |

*(Continued)*

**Table 2.** (Continued)

| Name | Formula | Description |
|---|---|---|
| Frequency standard deviation (F10) | $F10 = \sqrt{\sum_{i=1}^{N} (f_i - F8)^2 \times X_i / \sum_{i=1}^{N} X_i}$ | A measure of the dispersion or spread of the amplitudes across the frequency spectrum, indicating the variability of the signal's energy distribution. |

amplitude accelerates. Therefore, based on the three stages of overtravel change, samples can be categorized into three states: normal, mildly degraded, and severely degraded.

# 4 Feature selection and analysis

## 4.1 Feature selection model based on RFRS

Based on Table 2, it can be observed that time-frequency domain features share a common source of information, and some features are derived from mathematical transformations of other features. This implies that there is an overlap of information among the time-frequency domain features, which increases the complexity of data processing.

To better illustrate the application of RFRS in the domain of vibration signals for AC contactors, this section provides an analysis based on a specific example. The offline database is partitioned into a training set and a test set using a seven-fold cross-validation method, as shown in Fig 6. In each iteration, one of the seven subsets is used as the test set, while the remaining six subsets are used for training the RFRS.

Fig 7 illustrates the importance of each feature calculated after the introduction of the regularization term in RFRS. These features are then ranked in order of importance from highest to lowest, that is, {T5, T1, T3, T7, T6, F6,…}. Subsequently, the features are combined and the impact of these combinations on the model's overall accuracy is computed. Specifically, the first feature combination includes T5 alone, and the overall accuracy of the model is calculated; then, each subsequent feature is added to the combination one by one, forming new combinations, and the overall accuracy of the model at that point is computed. For instance, the second feature combination includes T5 and T1, and so on, where the $i$-th feature combination consists of the top $i$ important features, and the overall accuracy of the model under the influence of that combination is calculated. The relevant results are shown in Fig 8.

As can be observed from Fig 8, the overall accuracy of RFRS shows an increasing trend with the first six feature combinations. After the seventh feature combination, it reaches a relatively stable state, with fluctuations not exceeding 0.5%. This suggests that the information contained in the top six features, which are ranked by importance, may be sufficient to represent the state of the contactor. The features following the seventh feature may contain less information or be redundant with the first six features, and thus do not provide significant additional benefit to the model's performance.

Considering the potential for extreme values in the accuracy of state identification for individual states, this study undertakes a second round of judgment to assess whether the RFRS demonstrates extreme values or has achieved relative stability in accuracy for each state of the contactor. The process of this judgment is depicted in Fig 9. Specifically, the accuracy for the three states shows an increasing trend with feature combinations prior to the boundary defined by the 6th and 7th feature combinations, with the rise in accuracy exceeding 0.5%. Beyond this boundary, the accuracy for the three states associated with subsequent feature combinations remains relatively stable. At the same time, it can be noted that the RFRS has a higher recognition accuracy rate for State I and State III, both above 90%. However, for State

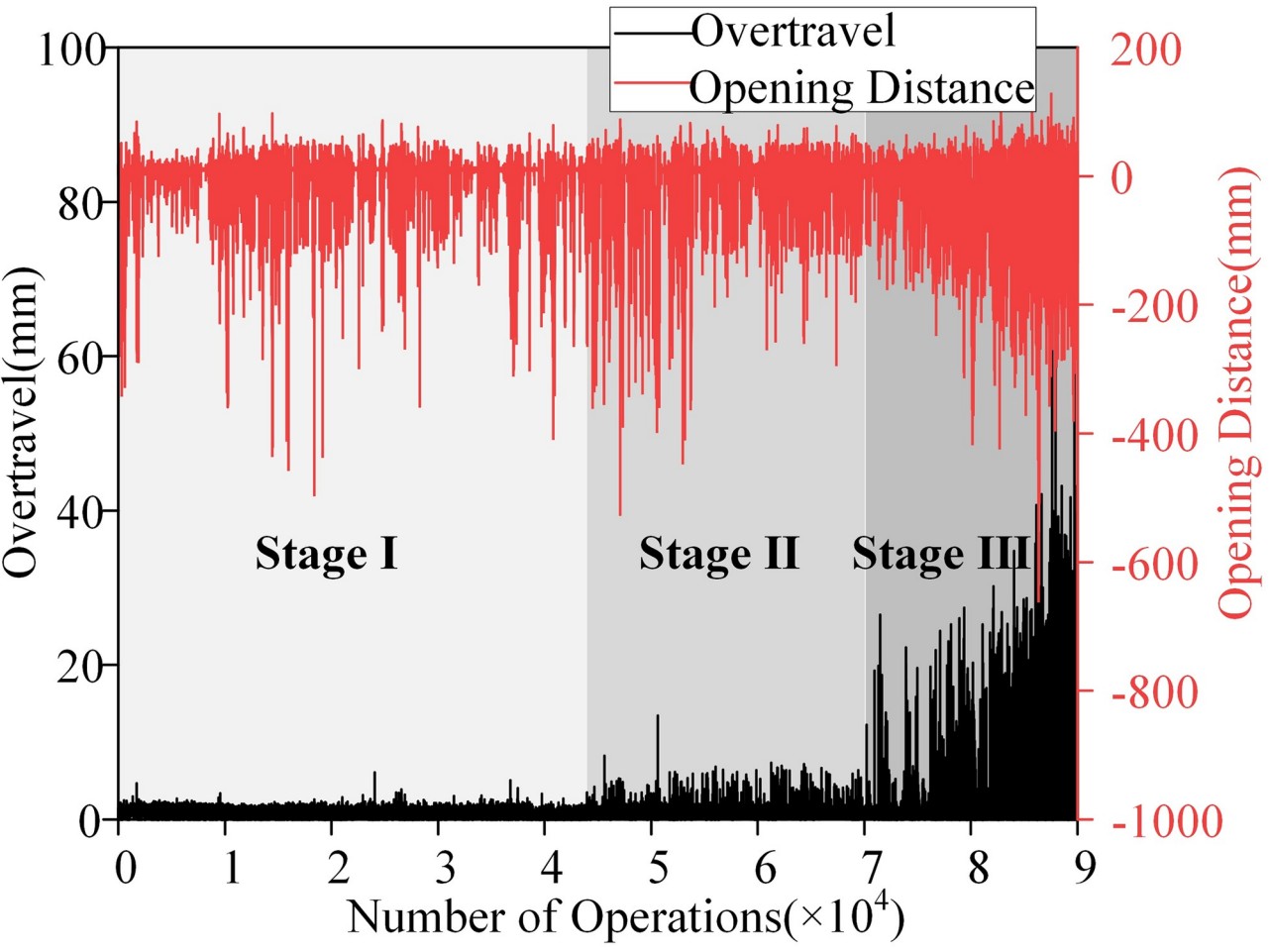

**Fig 5. Trends in overtravel and opening distance variations.**

II, the recognition accuracy rate of RFRS is lower, only reaching above 88%. In terms of the overall trend, the trend of accuracy rate changes for the three states is consistent with the trend of the overall accuracy rate. Using the same judgment criterion (a change in amplitude not exceeding 0.5%), this study considers that the information contained in the top six features, which are ranked by importance, is more critical. The information contained in the features after the 7th feature is less or redundant, hence the 6th round of feature combinations is taken as the optimal feature subset, which includes T1, T3, T5, T6, T7, and F6, with the results shown in Fig 10.

As depicted in Fig 10, the trends of variation for the key features selected by RFRS are presented. It can be observed that, with 40,000 operations as the demarcation point, before this point, the amplitude of time-domain features exhibits more pronounced changes, while the amplitude of frequency-domain features remains relatively flat. After the demarcation point, the amplitude changes of the time-domain features become more moderate, and there are no significant alterations in the amplitude of the frequency-domain features, except for a sudden spike at 90,000 operations. These distribution differences facilitate the accurate identification of Stage I and Stage III by RFRS. Concurrently, due to the subtle differences between Stage II

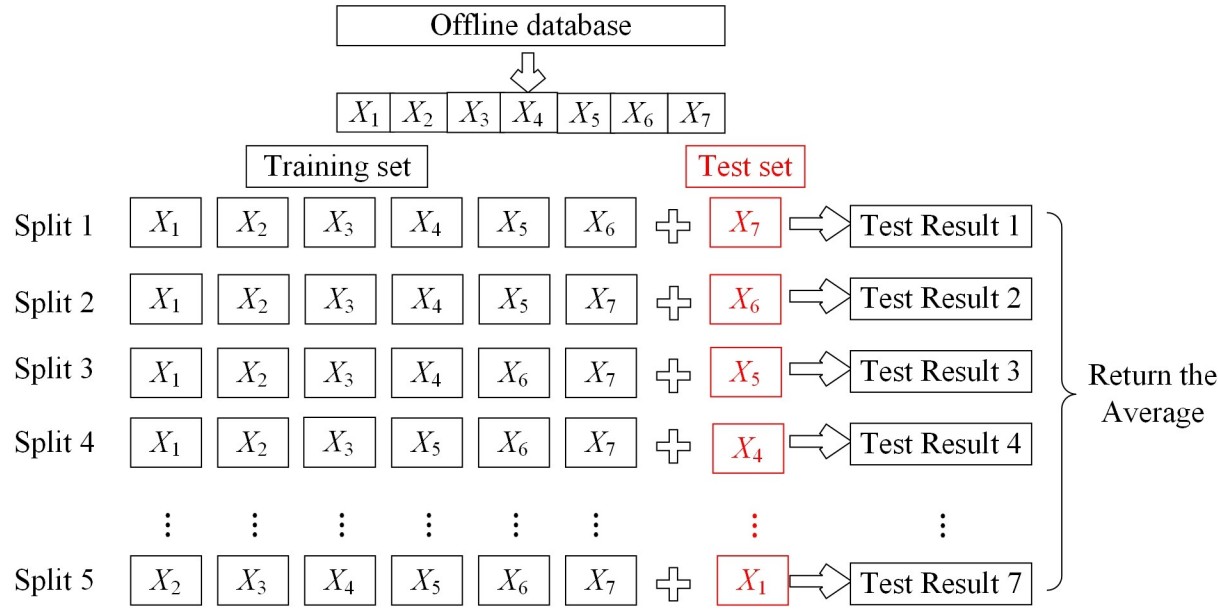

**Fig 6. Principle of sevenfold cross-validation.**

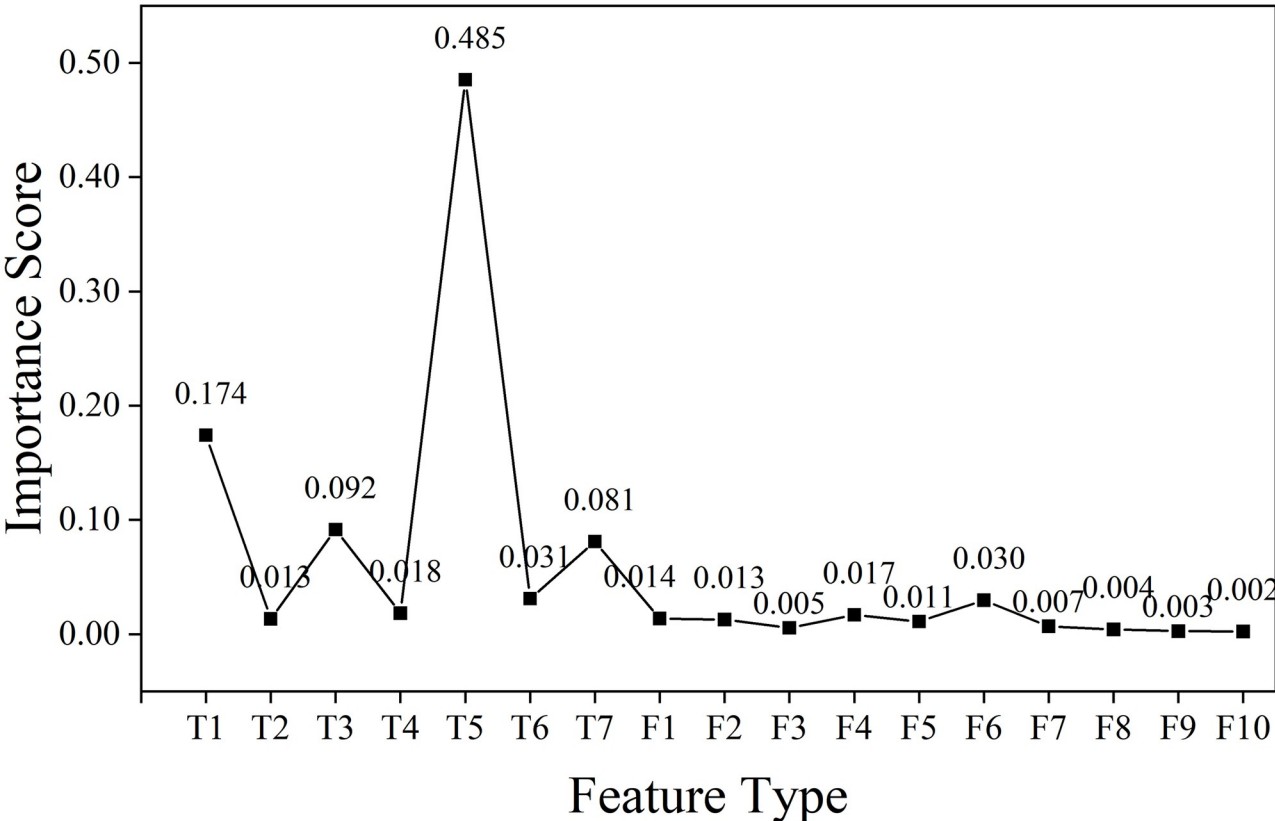

**Fig 7. Importance scores of time-domain and frequency-domain features.**

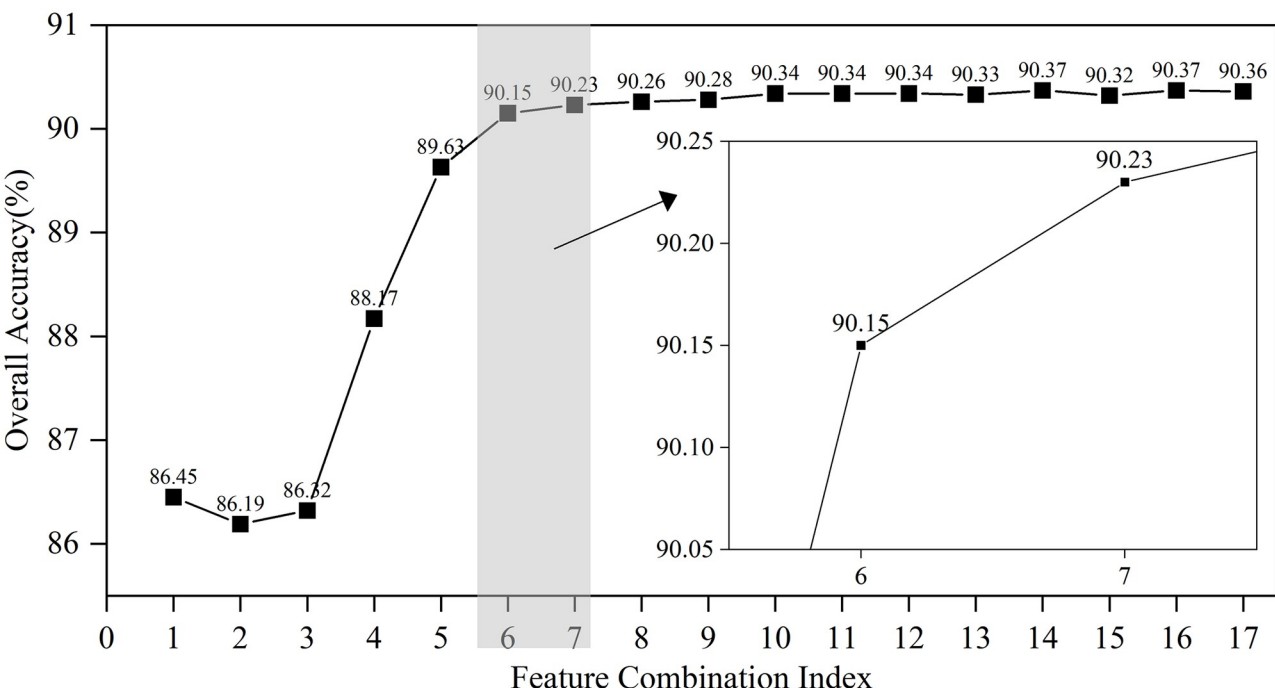

**Fig 8. Overall accuracy of various feature combinations.**

and Stage III, RFRS is unable to precisely identify Stage II, leading to an identification accuracy for Stage II that is only within the range of 88%-89%.

## 4.2 Comparative methods and evaluation metrics

To elucidate the advantages of RFRS in the feature selection of AC contactor vibration signals, this study has devised three comparative methods: Spearman's rank correlation coefficient (Spearman's rho), the filter method, and the embedded method. These methods possess strong interpretability and permit the intervention of subjective experience in the decision-making process, establishing them as prevalent feature selection techniques within the field of engineering technology.

(1) Spearman's rho is a non-parametric statistical measure used to evaluate the degree of correlation between two features. The coefficient ranges from -1 to 1, with higher absolute values indicating stronger correlations and greater redundancy in the information provided by the features. The formula for calculating this coefficient is as follows:

$$\rho_{\text{spearman}} = 1 - \frac{6\sum_{i=1}^{N} d_i^2}{N(N^2 - 1)} \tag{9}$$

In Eq (9), $d_i$ represents the measure between two features.

(2) The filter method is a variance-based feature selection technique that assesses the importance of features to a model by evaluating their variability across samples. Features with higher variance exhibit greater variability, indicating that they carry more effective

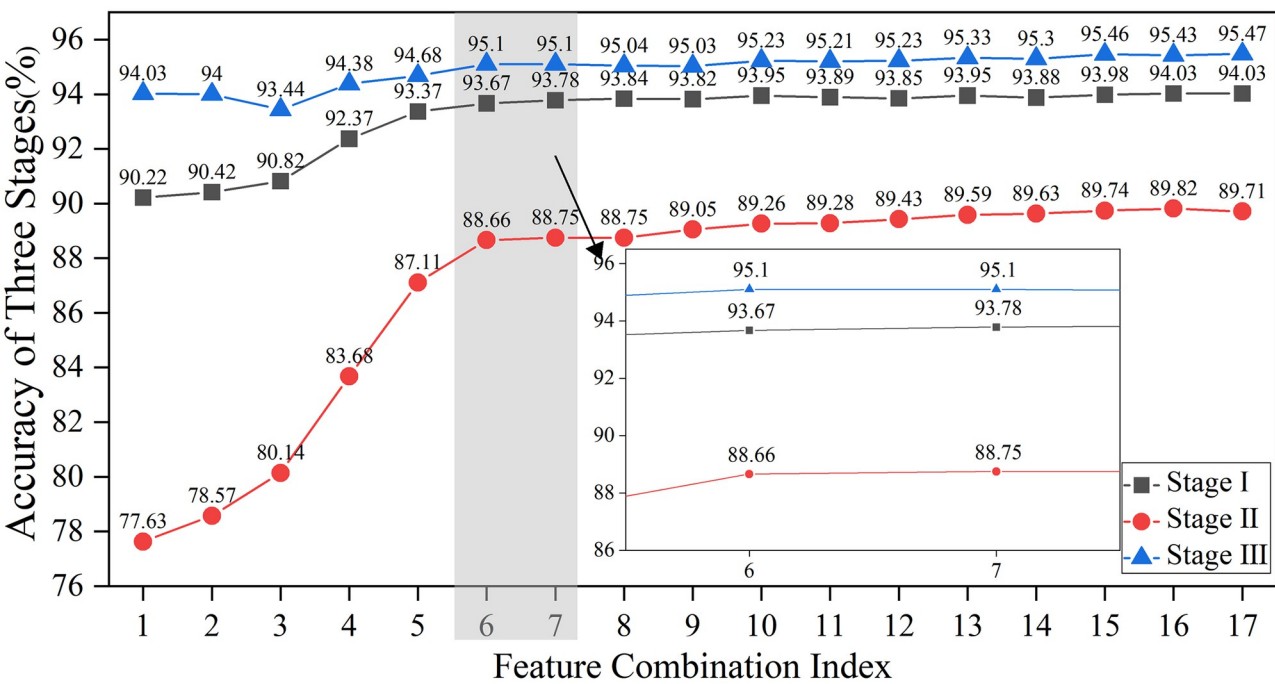

**Fig 9. Recognition accuracy for three stages across different feature combinations.**

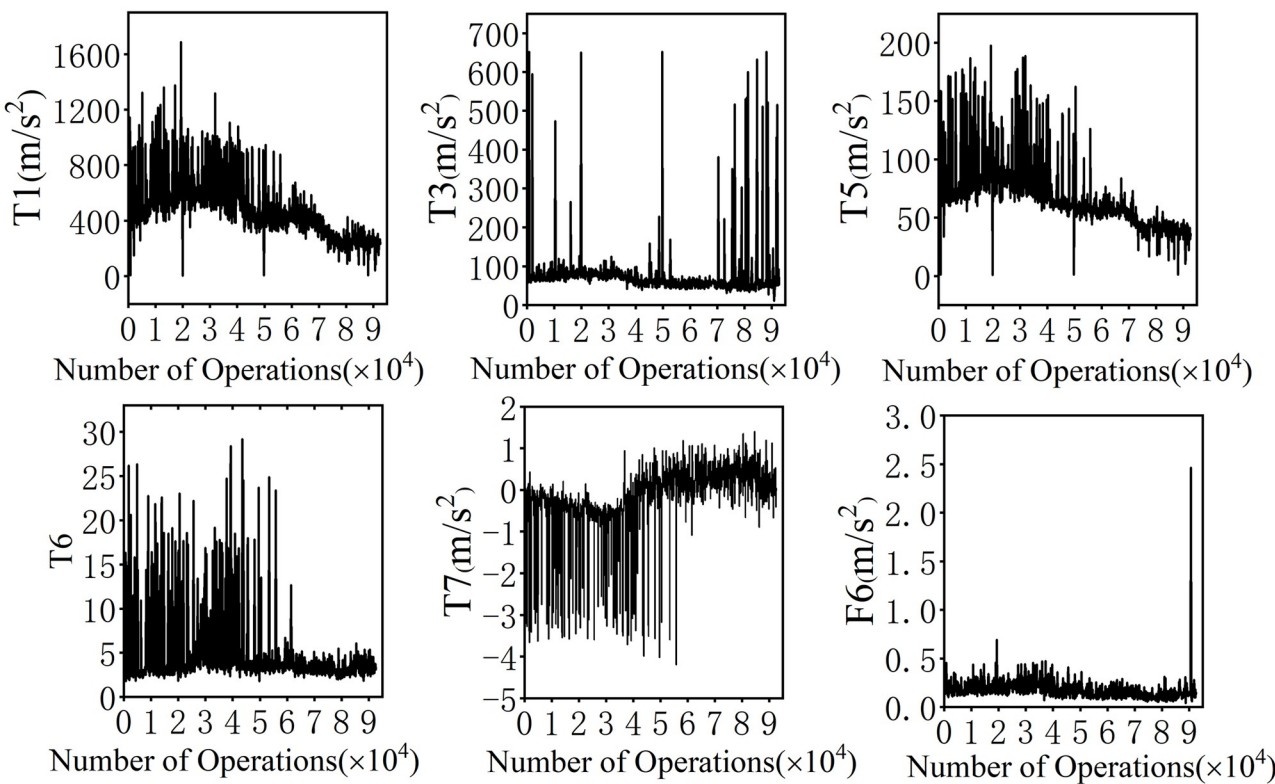

**Fig 10. Trend of key features.**

information; conversely, features with variances close to zero are eliminated due to their limited contribution of useful information. The variance is calculated using the following formula:

$$\sigma^2 = \frac{1}{N-1} \sum_{i=1}^{N} (x_i - \bar{x})^2 \tag{10}$$

In Eq (10), $\bar{x}$ is the sample mean of $x_i$.

(3) The embedded method is a RF-based feature selection technique that ascertains feature importance by evaluating the contribution of each feature in reducing impurity during the construction of decision trees. The importance is calculated using the following formula:

$$I_f = \frac{1}{T} \sum_{t=1}^{T} (C^{\mathrm{b}}_{\mathrm{Gain},t} - C^{\mathrm{a}}_{\mathrm{Gain},t}) \tag{11}$$

In Eq (11), $I_f$ denotes the importance of the $f$-th feature, $T$ is the total number of decision trees, and $C^{\mathrm{b}}_{\mathrm{Gain},t}$ and $C^{\mathrm{a}}_{\mathrm{Gain},t}$ represent the impurity gain before and after the split in the $t$-th tree, respectively.

To comprehensively evaluate the performance enhancement of different feature selection methods in the study of condition recognition, this study employs four types of classifiers and four commonly used evaluation metrics [23, 24].

The four types of classifiers are as follows:

(1) Light Gradient Boosting Machine (LGBM) is an optimized gradient boosting framework that demonstrates higher computational efficiency and superior performance compared to traditional tree models such as RF.

(2) K-Nearest Neighbors (KNN) is an instance-based learning method that classifies based on the distance between the test sample and the training samples, offering advantages in capturing complex boundaries and local structures.

(3) Support Vector Machine (SVM) is a classifier based on the principle of maximizing the margin, which achieves classification by finding the optimal hyperplane between data points and is commonly used for handling high-dimensional data and linearly separable problems.

(4) Logistic Regression (LR) is a fundamental model for multi-class problems, predicting probabilities through the output of the logistic function for classification, characterized by strong probabilistic interpretability.

The ensemble learning characteristics of LGBM, the distance-based classification of KNN, the margin maximization principle of SVM, and the probabilistic interpretability of LR encompass different learning paradigms, which can assess the applicability and effectiveness of feature selection methods from multiple perspectives.

The four commonly used evaluation metrics are as follows:

(1) Recall, a metric that quantifies the ratio of samples accurately classified by the model into a particular state relative to the total number of samples genuinely belonging to that state. Elevated recall signifies that the model exhibits heightened sensitivity in the detection of

specific degradation states. The calculation formula for this metric is as follows:

$$C_{\text{Recall}} = \frac{TP_y}{TP_y + FN_y} \tag{12}$$

(2) Precision, a metric that assesses the fraction of samples that genuinely belong to a designated class within the subset that the classifier has forecasted to fall under that class. Superior precision denotes that the classifier possesses a robust level of assurance in the prognostication of specific degradation states. The calculation formula for this metric is as follows:

$$C_{\text{Precision}} = \frac{TP_y}{TP_y + FP_y} \tag{13}$$

(3) F1-Score, representing the harmonic mean of precision and recall, encapsulates the classifier's aggregate performance pertaining to a particular class, notably in scenarios where a compromise between Precision and Recall is inevitable. The calculation formula for this metric is as follows:

$$C_{\text{F1-Score}} = 2 \times \frac{C_{\text{Precision}} \times C_{\text{Recall}}}{C_{\text{Precision}} + C_{\text{Recall}}} \tag{14}$$

(4) Accuracy, a metric that gauges the holistic efficacy of the classifier, offers a panoramic perspective on the classifier's ability to classify across all classes effectively. The calculation formula for this metric can be found in Eq (7).

Recall and precision respectively measure the sensitivity and reliability of a classifier, F1-Score is a comprehensive indicator, and accuracy provides a quick assessment of the classifier's overall recognition performance. Therefore, quantifying the performance of the classifier through these metrics can lead to a deeper understanding of the impact of different feature selection methods on the classifier.

## 4.3 Results and analysis

When various feature selection methodologies were applied to the selection of time-frequency domain characteristics of vibration signals from contactors, the outcomes were delineated in Table 3. Within Table 3, the embedded approach, constrained by the unbiasedness of RF, is unable to efficiently identify pivotal features. Conversely, RFRS employed a regularization factor to direct the split, thereby circumventing interference from minor gain directions, leading to an effective reduction in feature dimensionality. RFRS, in concordance with the Filter

**Table 3. Results of feature selection.**

| Feature selection method | Feature set | Selection result |
|:---:|:---:|:---:|
| RFRS | T1-T7, F1-F10 | T1,T3,T5,T6,T7,F6 |
| Spearman's Rho | | T1,T5,T6,T7,F6 |
| Embedded | | T1-T7, F1-F10 |
| Filter | | T1,T5,T7,F2,F10 |

**Table 4. Performance metrics on the training dataset.**

| Method | Classifier | Recall | F1-Score | Precision | Accuracy |
|---|---|---|---|---|---|
| RFRS | LGBM | 94.44% | 94.44% | 94.46% | 94.44% |
| | KNN | 92.31% | 92.29% | 92.30% | 92.29% |
| | SVM | 89.19% | 89.14% | 89.13% | 89.14% |
| | LR | 86.08% | 86.07% | 86.08% | 86.07% |
| | Average Value | 90.51% | 90.49% | 90.49% | 90.49% |
| Spearman's Rho | LGBM | 93.58% | 93.58% | 93.60% | 93.58% |
| | KNN | 91.28% | 91.26% | 91.27% | 91.28% |
| | SVM | 88.61% | 88.57% | 88.58% | 88.61% |
| | LR | 86.57% | 86.55% | 86.56% | 86.57% |
| | Average Value | 90.01% | 89.99% | 90.0% | 90.01% |
| Embedded | LGBM | 95.40% | 95.40% | 95.42% | 95.40% |
| | KNN | 92.14% | 92.14% | 92.16% | 92.14% |
| | SVM | 90.60% | 90.60% | 90.61% | 90.60% |
| | LR | 86.70% | 86.67% | 86.68% | 86.67% |
| | Average Value | 91.21% | 91.20% | 91.22% | 91.20% |
| Filter | LGBM | 91.84% | 91.81% | 91.84% | 91.84% |
| | KNN | 91.50% | 85.19% | 91.51% | 91.50% |
| | SVM | 89.44% | 89.40% | 89.41% | 89.44% |
| | LR | 85.74% | 85.69% | 85.67% | 85.74% |
| | Average Value | 89.63% | 88.02% | 89.61% | 89.63% |

method and Spearman's Rho, identified T1, T5, and T7 as the most critical vibration characteristics. Nonetheless, it is imperative to recognize the pronounced discrepancies among these three feature selection techniques concerning the determination of additional key features: RFRS deemed T3, T6, and F6 as pivotal; Spearman's Rho regarded T6 and F6 as significant; the Filter method nominated F2 and F10 as essential. To investigate the ramifications of these disparities on the research of state recognition, a seven-fold cross-validation approach is utilized to assess the efficacy of four distinct classifiers, with the findings detailed in Tables 4 and 5.

Table 4 presents the performance of various classifiers on the training dataset. When considering the average scores across the four metrics, the embedded method demonstrates superior learning capability for known samples compared to other approaches. However, this method fails to effectively select key features, instead incorporating all features into the learning process. On the premise of effectively reducing feature dimensionality, RFRS outperforms Spearman's Rho and the Filter in terms of learning ability for known samples.

Table 5 illustrates the performance of various classifiers on the test dataset. An integrated analysis of Tables 4 and 5 reveals that the embedded method, hindered by feature redundancy, exhibits inferior generalizability and lower recognition capability for unknown samples compared to RFRS and Spearman's Rho. Furthermore, Table 5 demonstrates that RFRS outperforms both Spearman's Rho and the Filter across all four evaluation metrics. Specifically, in terms of Recall, RFRS achieves an average of 87.37%, slightly higher than Spearman's Rho at 87.06%, and significantly better than the Filter, which stands at 85.52%. In Precision, RFRS shows an average of 88.38%, surpassing the 88.06% of Spearman's Rho and the 86.23% of the Filter, indicating its high efficiency in reducing false positives. The F1-Score reflects RFRS's balanced capability when considering both Recall and Precision, with an average F1-Score of 87.41%, outperforming the 87.10% of Spearman's Rho and the 85.58% of the Filter. Although RFRS has a slightly lower average accuracy at 85.74% compared to Spearman's Rho at 85.66%,

**Table 5. Performance metrics on the test dataset.**

| Method | Classifier | Recall | F1-Score | Precision | Accuracy |
|---|---|---|---|---|---|
| RFRS | LGBM | 90.78% | 90.72% | 91.70% | 88.52% |
| | KNN | 87.51% | 87.56% | 88.65% | 85.27% |
| | SVM | 87.49% | 87.57% | 88.47% | 86.45% |
| | LR | 83.69% | 83.78% | 84.71% | 82.73% |
| | Average Value | 87.37% | 87.41% | 88.38% | 85.74% |
| Spearman's Rho | LGBM | 90.18% | 90.14% | 91.09% | 88.19% |
| | KNN | 86.67% | 86.73% | 87.56% | 85.15% |
| | SVM | 87.27% | 87.33% | 88.17% | 85.95% |
| | LR | 84.12% | 84.20% | 85.40% | 83.33% |
| | Average Value | 87.06% | 87.10% | 88.06% | 85.66% |
| Embedded | LGBM | 88.69% | 88.66% | 89.92% | 88.69% |
| | KNN | 83.78% | 83.91% | 84.88% | 83.78% |
| | SVM | 87.07% | 87.15% | 87.82% | 87.07% |
| | LR | 83.77% | 83.83% | 84.48% | 83.77% |
| | Average Value | 85.83% | 85.89% | 86.78% | 85.83% |
| Filter | LGBM | 87.29% | 87.35% | 87.98% | 87.29% |
| | KNN | 85.11% | 85.19% | 85.82% | 85.11% |
| | SVM | 86.72% | 86.76% | 87.27% | 86.72% |
| | LR | 82.95% | 83.0% | 83.86% | 82.96% |
| | Average Value | 85.52% | 85.58% | 86.23% | 85.52% |

it still exceeds the Filter's 85.52%, with the differences being minimal, indicating that RFRS is not inferior in overall recognition performance. These results highlight the significant advantages of RFRS in the selection of time-frequency domain features for the vibration signals of AC contactors, providing an effective feature selection tool for condition recognition research.

## 5. Conclusion

This study proposes a feature selection method based on RFRS to address the issue of feature redundancy in the vibration signals of AC contactors. Through comprehensive analysis and experimental validation, the following conclusions have been drawn:

(1) This study introduces an enhancement to the traditional Random Forest algorithm by incorporating a regularization term to guide the direction of node splits and integrating recursive elimination concepts to optimize the stopping criteria, effectively reducing the dimensionality of time-frequency domain features.

(2) By comparing with three mainstream feature selection methods (Spearman's rank correlation coefficient method, the embedded method, and the filter method), and evaluating with various types of classifiers and multiple metrics, the results demonstrate that RFRS outperforms the comparative methods in overall performance. However, the superiority of RFRS is not absolute in specific contexts, and researchers must make decisions based on actual needs. For instance, when the objective is to minimize false judgments and the data is linearly separable, Spearman's rank correlation coefficient method may be more suitable.

(3) Conducting condition recognition research on AC contactors from the perspective of vibration signals still presents numerous challenges, particularly in terms of how to extract features with more pronounced trends from the vibration signals. Therefore, future work

will focus on two main aspects: one is the continuous expansion of the AC contactor dataset for testing and optimizing RFRS; the other is the exploration of novel feature extraction techniques, such as transforming data trends into image textures and utilizing image processing technologies to extract key features.

## Acknowledgments

We express our gratitude to the five authors acknowledged on the title page; their encouragement and trust were instrumental to the research presented herein

## Author Contributions

**Data curation:** Xinzhi Qi.

**Funding acquisition:** Shuxin Liu.

**Methodology:** Shuxin Liu, Xinzhi Qi, Chaojian Xing, Xianfeng Lv.

**Software:** Xin Ming.

**Supervision:** Xin Ming.

**Validation:** Xin Ming.

**Visualization:** Xianfeng Lv.

**Writing – original draft:** Shuxin Liu, Xinzhi Qi.

**Writing – review & editing:** Shuxin Liu, Xinzhi Qi.

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
