## [Decision Letter · Decision Letter 0]

4 Jul 2024

PONE-D-24-16516Research on Feature Selection for AC Contactor Vibration Signals Based on Regularized Random Forest with Recursive SelectionPLOS ONE

Dear Dr. Qi,

Thank you for submitting your manuscript to PLOS ONE. After careful consideration, we feel that it has merit but does not fully meet PLOS ONE’s publication criteria as it currently stands. Therefore, we invite you to submit a revised version of the manuscript that addresses the points raised during the review process.

We look forward to receiving your revised manuscript.

Kind regards,

Caio Bezerra Souto Maior, Ph.D.

Academic Editor

PLOS ONE

Journal Requirements:

https://journals.plos.org/plosone/s/file?id=ba62/PLOSOne_formatting_sample_title_authors_affiliations.pdf"

3. Thank you for stating the following financial disclosure: "This work was supported by National Natural Science Foundation of China (No. 51977132); Liaoning Science and Technology Major Project (2020JH1/10100012); Shenyang Young and Middle-aged Science and Technology Innovation Talent Program (RC210354)".

4. Thank you for stating the following in the Acknowledgments Section of your manuscript: "This work was supported by National Natural Science Foundation of China (No. 51977132); 

Liaoning Science and Technology Major Project (2020JH1/10100012); Shenyang Young and Middleaged Science and Technology Innovation Talent Program (RC210354)".

Please remove any funding-related text from the manuscript and let us know how you would like to update your Funding Statement. Currently, your Funding Statement reads as follows: "This work was supported by National Natural Science Foundation of China (No. 51977132); Liaoning Science and Technology Major Project (2020JH1/10100012); Shenyang Young and Middle-aged Science and Technology Innovation Talent Program (RC210354)."

5. In the online submission form, you indicated that "Due to the privacy nature of the data, it cannot be made publicly available. However, the data can be accessed by any interested party upon request to the corresponding author. Please contact qixinzhi_1999@163.com for further information regarding data access."

Reviewers' comments:

Reviewer's Responses to Questions

**Comments to the Author**

1. Is the manuscript technically sound, and do the data support the conclusions?

Reviewer #1: Yes

Reviewer #2: Partly

2. Has the statistical analysis been performed appropriately and rigorously? 

Reviewer #1: Yes

Reviewer #2: No

3. Have the authors made all data underlying the findings in their manuscript fully available?

Reviewer #1: Yes

Reviewer #2: No

4. Is the manuscript presented in an intelligible fashion and written in standard English?

Reviewer #1: Yes

Reviewer #2: Yes

5. Review Comments to the Author

Reviewer #1: The paper introduces an approach for feature selection using a regularized random forest and a recursive selection procedure, but with the traditional regularization term being modified to include the ratio of importance, relative to the maximum importance among all features, of a certain feature in the penalization. A total of 7 time-domain and 10 frequency-domain features are analyzed and selected, via both traditional feature selection methods and the proposed one, and then evaluated using four classifiers. The paper’s results show some advantage of the proposed method over other traditional feature selection methods. Some comments:

1) Review of acronyms’ definitions along all paper. Many acronyms are defined multiple times, which is not correct. For example, the “(RFRS)” acronym definition appears in the abstract, in the keywords, in the last paragraph of the introduction and then twice in the Conclusion section. Other acronyms are also defined multiple times, such as “(RF)”. Please revise those acronyms over the entire paper and only define them the first time they appear.

2) 3.3.1 Feature Extraction: pages 8-9 include the name, description and formula of multiple features, organized per paragraph. This information would be better suited in a table with three columns: name, description and formula.

3) Sections 3.3.1 and 3.3.2 have the same name “Feature Extraction”. I also believe section 3.3.2 is related to experiment and/or some sort of data processing, not Feature Extraction itself, so this name does not seem not suitable unless extra information is provided in the section’s title.

4) Section 3.3.2 Feature Extraction: in the last paragraph of page 9, Stage I is stated as # of operations ranging from 10000 to 30000 and Stage II from 30000 to 70000. In the first paragraph of page 10, Stages I and II are stated as from 10000 to 40000 and 40000 to 70000 instead. Which one is the correct value? The first time (page 9) the stages are cited must be adjusted to the correct value, while the second time (page 10) should not include the parenthesis part after ‘Stage X’ as the range for each stage was already described in the previous paragraph.

5) Fig 8: I believe this graph is relative to the three Stages described in section 3.3.2, but its legend uses the term ‘Phase X’ instead, mentioned nowhere in the paper. Either describe the term along the paper, if it is supposed to be Phase, or change it to ‘Stage X’ instead.

6) Table 4 includes the metrics Recall, F1-Score and Precision for the four feature selection methods analyzed and for four different classifiers. Accuracy should also be included in the table. Although generic and misleading in some cases, it is still an easy-to-interpret metric that, alongside the others already provided by the authors, helps understanding the advantages of the proposed approach.

7) 4.3 Results and Analysis: some more robust classifiers could be evaluated, such as MLP and a traditional Random Forest classifier.

8) 5. Conclusion: the last sentence of topic (3) is almost the same as the sentence that appears in the last paragraph of Conclusion section. Topic (3) is described by the authors as a ‘future steps’ guideline, therefore the sentence “In summary, the RFRS-based feature selection…” does not fit there. Also, future steps were not correctly defined: “exploration of new feature extraction techniques”, an example of such new techniques could be added. Also, testing your proposed methodology in different AC contactors’ datasets is an important future step and could be added in this topic.

Reviewer #2: The paper introduces a feature selection strategy based on Regularized Random Forest with Recursive Selection applied to vibration signals of AC contactors to enhance the accuracy of their condition identification. The topic is interesting and worth investigating. However, I suggest the following revision points before the paper is considered for publishing.

1) In the abstract, there is no contextualization of the problem. Why is this development worth investigating?

2) In the abstract, what is the contribution of the paper?

3) In the introduction, please consider reformulating or removing the phrase “It is known from statistical knowledge that the standard deviation is closely related to variance” given that “standard deviation is the square root of the variance”.

4) In the introduction, “[19-27]” avoid using nested citation.

5) In equation (1), what is wr and wl? Please define all terms in all equations.

6) Fig. 2 is not cited in the text.

7) Regarding the figure order, I suggest introducing and citing the figure, inserting the figure in the manuscript and then discussing it. Normally, the figures appear with any discussion.

8) Fig. 3 and its font are small, please increase it to improve readability.

9) Fig. 4 and its font are small, please increase it to improve readability.

10) In Section 3.2, few details are given about the vibration time series. It should be provided and discussed the sampling frequency in Hz, the length of the time waveform, etc? Was the vibration measurement triggered with the AC contactor movement? Each measurement contains how many openings and closings of the contactor?

11) In Table 1, It is not clear if the information presented refers to the accelerometer, contactor or experiment. Please clarify it in the table and text.

12) Avoid using “frequency time waveform”, I suggest using frequency spectrum instead.

13) In equation (22), “fi refers to the frequency-domain value” explain it better.

14) In Section 4.1, how training and testing datasets are selected? How many contactors have been used? How long did it take to carry out measurements? How many hours/days of experiment have been carried out? What has been done to avoid overfit of the models? What are the results during training and testing? How many samples have been considered in the training and testing datasets?

15) I understood that four methods have been used to compare the results: RFRS, Spearman's Rho, Embedded and Filter. I suggest describe specifically each method, considering the methods’ name used in the manuscript in a specific section. It is not clear what are the specificities of each method.

16) In Section 4.3, why were the models LGBM, KNN, SVM, and LR, chosen? It is important to include a brief discussion about them and the reason to choose them.

17) In Section 4.3, why were the KPIs, f recall rate, F1-Score, and precision, chosen? It is important to include a brief discussion about them the reason to choose them.

18) In Table 4, I suggest underlining the best results for each classification model.

19) In “As depicted in Table 4, the average scores for Recall, F1-Score, and Precision for RFRS are 87.37%, 87.41%, and 84.71%, respectively; for Spearman's rho, the corresponding scores are 82.06%, 87.1%, and 88.06%, respectively; and for the Filter method, the scores are 85.52%, 85.58%, and 86.23%, respectively. Comparatively, the Recall score of RFRS is 5.31% higher than that of Spearman's rho and 1.85% higher than the Filter method; its F1-Score is 0.31% higher than Spearman's rho and 1.83% higher than the Filter method; whereas its Precision score is 3.35% lower than Spearman's rho and 1.52% lower than the Filter method.”, the average values are not highlighted in the Table.

In the Conclusion, “current mainstream feature selection methods.”, please name the methods. In Conclusion, “, although the specific application scenarios still require researchers to make decisions based on actual needs.”, please be more specific.

6. PLOS authors have the option to publish the peer review history of their article (what does this mean?). If published, this will include your full peer review and any attached files.

Reviewer #1: No

Reviewer #2: No

---

## [Author Response · Author response to Decision Letter 0]

24 Jul 2024

We would like to express our gratitude for the opportunity to revise our manuscript. Please find the "Response to Reviewers" document attached, which includes our detailed responses to each of the specific comments made by the reviewers and the editor. Due to the extensive length and inclusion of figures, we have prepared this comprehensive document to ensure a thorough and clear communication of our revisions.

In the "Response to Reviewers" document, we have addressed the feedback point by point, providing explanations and evidence of the changes made in the revised manuscript. Additionally, we have included any necessary figures and tables to illustrate our responses where appropriate.

We trust that this approach will facilitate a smooth review process and have confidence that our revisions meet the high standards of your esteemed journal.

---

## [Decision Letter · Decision Letter 1]

26 Aug 2024

Research on feature selection for AC contactor vibration signals based on regularized random forest with recursive selection

PONE-D-24-16516R1

Dear Dr. Qi,

We’re pleased to inform you that your manuscript has been judged scientifically suitable for publication and will be formally accepted for publication once it meets all outstanding technical requirements.

Kind regards,

Caio Bezerra Souto Maior, Ph.D.

Academic Editor

PLOS ONE

Additional Editor Comments (optional):

Reviewers' comments:

Reviewer's Responses to Questions

**Comments to the Author**

1. If the authors have adequately addressed your comments raised in a previous round of review and you feel that this manuscript is now acceptable for publication, you may indicate that here to bypass the “Comments to the Author” section, enter your conflict of interest statement in the “Confidential to Editor” section, and submit your "Accept" recommendation.

Reviewer #1: All comments have been addressed

Reviewer #2: All comments have been addressed

2. Is the manuscript technically sound, and do the data support the conclusions?

Reviewer #1: Yes

Reviewer #2: Yes

3. Has the statistical analysis been performed appropriately and rigorously? 

Reviewer #1: Yes

Reviewer #2: Yes

4. Have the authors made all data underlying the findings in their manuscript fully available?

Reviewer #1: Yes

Reviewer #2: Yes

5. Is the manuscript presented in an intelligible fashion and written in standard English?

Reviewer #1: Yes

Reviewer #2: Yes

6. Review Comments to the Author

Reviewer #1: (No Response)

Reviewer #2: Thanks for the authors' effort to address all my comnents. I consider the paper as accepted. However, I still have two comments:

1. In Table 1, include the units between parentheses and not with "/".

2. The unit is not MHz/s.but MHz.

7. PLOS authors have the option to publish the peer review history of their article (what does this mean?). If published, this will include your full peer review and any attached files.

Reviewer #1: No

Reviewer #2: No

---

## [Editor Report · Acceptance letter]

28 Aug 2024

PONE-D-24-16516R1 

PLOS ONE

Dear Dr. Qi, 

I'm pleased to inform you that your manuscript has been deemed suitable for publication in PLOS ONE. Congratulations! Your manuscript is now being handed over to our production team.

Kind regards, 

on behalf of

Dr Caio Bezerra Souto Maior 

Academic Editor

PLOS ONE